# Effect of Site and Phenological Status on the Potato Bacterial Rhizomicrobiota

**DOI:** 10.3390/microorganisms10091743

**Published:** 2022-08-29

**Authors:** Lisa Cangioli, Marco Mancini, Ada Baldi, Camilla Fagorzi, Simone Orlandini, Francesca Vaccaro, Alessio Mengoni

**Affiliations:** 1Department of Biology, University of Florence, Via Madonna del Piano 6, 50019 Sesto Fiorentino, Italy; 2Department of Agriculture, Food, Environment and Forestry (DAGRI), University of Florence, Piazzale delle Cascine 18, 50144 Firenze, Italy

**Keywords:** potato, rhizosphere, microbiota, ecosystem functions

## Abstract

The potato is the fourth major food crop in the world. Its cultivation can encounter problems, resulting in poor growth and reduced yield. Plant microbiota has shown an ability to increase growth and resistance. However, in the development of effective microbiota manipulation strategies, it is essential to know the effect of environmental variables on microbiota composition and function. Here, we aimed to identify the differential impact of the site of cultivation and plant growth stage on potato rhizosphere microbiota. We performed a 16S rRNA gene amplicon sequencing analysis of rhizospheric soil collected from potato plants grown at four sites in central Italy during two phenological stages. Rhizomicrobiota was mainly composed of members of phyla Acidobacteriota, Actinobacteriota, Chloroflexi, and Proteobacteria and was affected by both the site of cultivation and the plant stages. However, cultivation sites overcome the effect of plant phenological stages. The PiCRUST analysis suggested a high abundance of functions related to the biosynthesis of the siderophore enterobactin. The presence of site-specific taxa and functional profiling of the microbiota could be further exploited in long-term studies to evaluate the possibility of developing biomarkers for traceability of the products and to exploit plant growth-promoting abilities in the native potato microbiota.

## 1. Introduction

The potato is the fourth major food crop in the world, and its cultivation in the field can encounter many problems that result in poor growth and reduced yield [1]. Aside from plant genetics, crop yield is the result of a variety of factors, including agronomic practices, soil, climatic conditions, and plant microbiota [2]. The plant microbiota, and especially the rhizospheric microbiota, has been clearly related to crop yield and resistance to biotic and abiotic stresses [3,4,5,6,7]. As for many other plant species (see, for instance, [8]), the microbiota of potato rhizosphere has been shown to be strongly influenced by the plant developmental stage [9,10,11,12], with the flowering stage generally having the highest diversity [9,10]. However, the effect of the site of cultivation is also relevant, since pedoclimatic conditions affect both soil microbiota composition and plant growth. In a survey of rhizosphere soil samples from potatoes grown at different altitudes [9], it was shown that richness (alpha diversity) was more affected by the site of sampling, as conditioned by different soil properties and climatic conditions, while the taxonomic composition was more significantly affected by the plant vegetation stage. When considering the exploitation of the plant microbiota for improving plant growth, in terms of the preparation and formulation of bioinoculants, it is vital to identify the core set of microbial taxa associated with the target plant species—i.e., those which are always present, irrespective of environmental changes (e.g., the site of cultivation, growth stage, agronomic practices) and, possibly, of plant variety [13]. At the same time, the identification of taxa that can vary depending on the above-mentioned variables can permit the design of tailored bioinoculant formulations that are optimized under specific conditions [14,15]. It can also define possible microbiological markers of, for instance, the locality of cultivation, to guarantee traceability and enhance the area of the origin of the product [16,17,18]. This latter aspect is particularly relevant for Italy. In fact, although Italy is not one of the main potato producers [19], there are several traditional dishes that use local varieties of potatoes—especially sites suitable for potato cultivation [20,21,22] in the hilly and mountainous areas of the Central Apennines.

Consequently, it is highly relevant to perform an investigation of potato-associated microbiota to provide additional data for a better understanding of the role phenological stages and cultivation sites play on potato microbiota diversity and, consequently, future rational development of bioinoculants and microbiome fingerprinting methods.

In this work, we aimed to identify the components of the bacterial rhizospheric microbiota of potatoes and their changes in relation to the site of cultivation and the plant growth stage on cv. Kennebec, a classical potato variety grown in the Italian Central Apennine [23]. We answered this question by performing a 16S rRNA gene amplicon sequencing analysis of rhizospheric soil collected from sites differing in pedoclimatic conditions during two phenological stages: flowering and harvesting.

## 2. Materials and Methods

### 2.1. Potato Field Selection, Sampling, and Physicochemical Analysis

Four sampling sites cultivated with potatoes (*Solanum tuberosum* L.), cv. Kennebec, were selected in Tuscany (central Italy) at different altitudes (Table 1). For the locality of Gavigno, two sampling sites were considered for the evaluation of differences over a small distance (ca. 650 m distance). Fields were tilled and fertilized (Table 1) before sowing. and sites did not receive additional irrigation. For each sampling site, three soil samples were collected at the time of sowing, at a depth of 15 cm, and then mixed and analyzed to assess the main physical and chemical properties. The climatic conditions were computed for the entire growing cycle, from approximately early April 2020 (sowing) to late August 2020 (maturity). The temperature pattern was computed using growing degree days (GDDs) following the method suggested by [24]:(1)$$\sum \mathrm{GDD}\;=\frac{\left(\mathrm{Tmax}\;+\;\mathrm{Tmin}\right)}{2}-\mathrm{Tbase}$$
where ∑GDD is the sum of the degree days, Tmax is the maximum air temperature (°C), Tmin is the minimum air temperature (°C), and Tbase is the lower baseline temperature of the crop (°C). In this work, a Tbase of 4.4 °C was used [25]. The amount of water supplied by rainfalls was calculated as cumulated monthly precipitation.

The sampling of soil was performed during the flowering and harvesting stages, from June to August 2020, under the guidance of the sampling campaign for the Crop Microbiome Survey (https://www.globalsustainableagriculture.org/the-crop-microbiome-survey/) [26], following the protocol defined in March 2020 (https://www.globalsustainableagriculture.org/wp-content/uploads/CropMicrobiome_March-2020.pdf, accessed on 1 June 2020). For each sampling site and time, six samples were prepared, for a total of 48 samples (Appendix A). Each sample was constituted of five individual plants at the flowering stage or the harvesting stage. Selected plants were at a distance of at least 5 m from each other. After digging up the entire plant (10 cm radius around the plant and down to 20 cm soil depth) and shaking off the plant roots, the soil remaining attached to the roots was detached with a sterile scalpel and considered to be rhizospheric soil. Rhizospheric soils from each of the five plants were mixed, vortexed, and stored at −80 °C until DNA extraction. Physicochemical analyses of the soils were performed as reported previously [27].

### 2.2. eDNA Extraction and 16S rRNA Gene Amplicon Sequencing

Environmental DNA (eDNA) was extracted from 500 mg of soil using the DNeasy PowerSoil Pro (Qiagen Italy, Milan, Italy). From the extracted DNA, the bacterial V4 region of the 16S rRNA genes was amplified, as reported previously, from 1 ng of total DNA [27]. Primers 515F (5′-GTGCCAGCMGCCGCGGTAA-3′) and 806R (5′-GACTACHVGGGTATCTAATCC-3′) were used [28]. Reactions were performed in a 25 µL total volume with KAPA HiFi HotStart ReadyMix, (Kapabiosystems, Cape Town, South Africa) 1 µM of each primer with 25 cycles, with the following temperature profile: 30″ 95 °C, 30″ 55 °C, and 30″ 72 °C. PCR products were sequenced in a single run using Illumina MiSeq technology with a pair-end sequencing strategy and a MiSeq Reagent Kit v3 (Illumina, San Diego, CA, USA). Library preparation (Nextera XT, Illumina, San Diego, CA, USA) and demultiplexing were performed following Illumina’s standard pipeline, as previously reported [29].

### 2.3. Raw Data Processing, Clustering, and Taxonomic Assignment of Reads

Illumina reads were trimmed using the ‘Trim Galore!’ tool on the Galaxy server (https://usegalaxy.org/, accessed on 10 October 2021). Paired-end sequences were clustered into amplicon sequence variants (ASVs), following the DADA2 pipeline (version 1.16) [30]. Sequences were filtered and dereplicated using default parameters (MaxN = 0, truncQ = 2, rm.phix = TRUE, and maxEE = 2) to collapse identical reads into unique sequences, and chimeras were removed. The taxonomy assignment was carried out by comparing our data against the SILVA NR99rel138 standard database of bacteria [31], using the ‘DECIPHER’ R package (version 2.18.1) [32] as an implementation of DADA2 (SSU version 138 available at: http://www2.decipher.codes/Downloads.html, accessed on 20 October 2021). Annotated ASV count tables were processed using the Phyloseq package in R environment version 4.0.5 [33].

### 2.4. Diversity of the Rhizosphere Microbiota

The analysis of the microbial communities was mainly performed by following a previously published pipeline [27] through the ‘Phyloseq’ R package (version 1.34.0) [33]. For alpha diversity analyses, the Shannon and Simpson indices were calculated and plotted using the function ‘diversity()’ within the ’microbiome’ R package (version 1.12.0) [34]. Good’s coverage and evenness indices were calculated through the R functions ‘goods()’ and ‘evenness()’, respectively, within the ‘microbiome’ R package (version 1.12.0). Two evenness indices were computed (Camargo’s index [35] and Pielou’s index [36]). Rarefaction curves were generated using the ‘ggplot2′ (version 3.3.3) [37] and ‘ranacapa’ (version 0.1.0) [38] R packages by using the ‘ggrare’ function on the phyloseq object. The ‘ggplot2′ R package (version 3.3.3) was used to generate relative abundance plots. Wilcoxon tests for multiple comparisons of averages were performed on alpha diversity indices using ’ggviolin()’ and ‘stat_compare_means()’ within the ’ggpubr’ R package (version 0.4.0).

### 2.5. Effect of Site and Phenological Stage on Rhizosphere Taxonomic Composition

For the ordination plots of the phyloseq objects, a multivariate analysis based on the Bray–Curtis distance and nonmetric multidimensional scaling (nMDS) ordination (generated using the “ordinate” function) was performed, and plots were generated using the “plot_ordination()” function within the phyloseq package. Different community structures were analyzed using permutational a multivariate analysis of variance (PERMANOVA), which was performed using the R packages ‘ggplot2′ (version 3.3.3), ‘vegan’ (version 2.5-7), and ‘pairwise.Adonis’ (version 0.0.1), using the functions ’adonis2()’ and ’pairwise.adonis()’, respectively.

A differential abundance analysis was carried out using the R package DeSeq2 v1.30.1 [39], in order to identify the ASVs/taxa differentially expressed along the samples. The functions ‘phyloseq_to_deseq2()’, ‘DESeq()’, and ‘results()’ were used.

### 2.6. Functional Inference on the Rhizosphere Microbiota

The metabolic pathway abundances and functional gene profiles were obtained by running the software PICRUSt2 (Phylogenetic Investigation of Communities by Reconstruction of Unobserved States, https://github.com/picrust/picrust2, accessed on 20 October 2021), as previously reported in [27], using the Galaxy pipeline of the Deng Lab [40]. The ranking of the most relevant pathways for variety differentiation was conducted by running a SIMPER test using the function ‘simper()’ with the ‘vegan’ R package (version 2.5-7).

## 3. Results

### 3.1. Pedoclimatic Conditions

Results of physical and chemical analyses of the soil are reported in Appendix A. Based on the USDA soil textural triangle classification system [41], all the collected soil belongs to the silty loam class, except Gavigno 2 which is classified as loam. The soils of Gavigno, being located in a wooded area, are richer in organic matter and have a more acid pH than the soils of the Val di Chiana (hilly area) and Prato (urban area). The soils mostly differed in P_2_O_5,_ K_2_O_,_ Mg, and Ca contents, and all samples showed different groupings when subjected to a principal component analysis (PCA) performed on the soil physicochemical parameters (Figure 1).

Climatic conditions are summarized in Appendix A. Significant variability between the timepoints (in months) and sampling sites was observed for both temperature and precipitation. The monthly GDD varied from 172.5 (April) to 511.15 (August), 219.3 to 606.25, and 327.35 to 694.2 for Gavigno, Val di Chiana, and Prato, respectively. As expected, Prato was the warmest site, with 2686.25 GDD, followed by Val di Chiana (2211.65 GDD) and Gavigno (1806.4 GDD). Gavigno was the wettest sampling site, reaching a cumulative precipitation level of 499 mm for the entire growing season, with maximum precipitation (149.2 mm) recorded in June and minimum precipitation (43.0 mm) recorded in July. The same trend was also observed in Val di Chiana, with maximum (105.2 mm) and minimum (14.2 mm) rainfalls recorded in June and July, respectively. Prato was the driest site, reaching a cumulative precipitation level of 239.2 mm from April to August, which is less than half of the levels found in Gavigno. Unexpectedly, August was the wettest month (70.2 mm), while July was confirmed as the least rainy month (22 mm).

### 3.2. Overall Diversity of the Rhizosphere Bacterial Microbiota

A total of 3′760′646 16S rRNA reads were obtained, and 2′721′248 (72% of total reads) passed quality filtering (Appendix A). After the clustering step, a total of 14′332 ASVs were obtained (Dataset S1). Rarefaction curves obtained from the ASVs reached a plateau for all samples (Appendix A), indicating a satisfactory survey of the bacterial diversity (Good’s coverage, Appendix A).

### 3.3. Location Is the Main Driver of the Taxonomic Composition of Potato Bacterial Rhizomicrobiota

Alpha diversity estimates were obtained, and comparisons between location and phenological stages were performed. Wilcoxon tests indicated that the Simpson index and both evenness indices showed different values among localities (Figure 2, Appendix A). Moreover, the PERMANOVA reported that the phenological stage, and the interaction between locality and stage, significantly contributed to samples differences (Table 2, Appendix A).

The same pattern was observed for the taxonomic similarities among samples (Figure 3), with the two Gavigno sets of samples (Gavigno 1 and Gavigno 2) grouping together and remaining separated from the Prato and Val di Chiana samples. Within the same locality, the two phenological stages showed a separation. A PERMANOVA test (Appendix A) confirmed the high relevance of localities (sites, F = 1.64, *p* < 0.001) but also confirmed the contribution of the phenological stage (F = 1.17, *p* < 0.01) and their interaction (F = 1.19, *p* < 0.001) in the sample differences.

### 3.4. Taxonomic Signatures of Localities and Phenological Stages

Figure 4 reports the relative abundances of taxonomies at the phylum level. The analysis of the taxonomic composition revealed that the most abundant phyla were Acidobacteriota, Actinobacteriota, Chloroflexi, and Proteobacteria.

In order to check for the presence of the differential abundance of taxonomies with respect to localities and phenological stages, the package DESeq2 was used in the overall dataset. While no single ASV was found to be statistically different in abundance, with respect to the phenological stage for all samples, 15 ASVs were found to be differentially distributed among localities—9 of which were affiliated with the class Actinobacteria (Appendix A), suggesting the presence of soil-/locality-related microbiota signatures.

### 3.5. Functional Potential of the Potato Rhizomicrobiota

Functional profiles were inferred from retrieved taxonomies, in terms of the metabolic pathways present in the bacterial taxa of the overall potato rhizomicrobiota. We identified a total of 434 pathways distributed in all samples (Appendix A). Using a SIMPER test, we identified nine and three pathways that significantly differentiated samples with respect to locality and phenological stage, respectively (Figure 5). The pathways included abilities related to the biosynthesis of molecules (ENTBACSYN-PWY, enterobactin biosynthesis), the degradation of plant metabolites (PWY-5266, p-cymene degradation; PWY-5273, p-cumate degradation), anaerobic metabolism (PWY0-1533, methylphosphonate degradation; PWY-7007, methyl ketone biosynthesis; PWY-6876, isopropanol biosynthesis), carbon fixation (PWY-5392, reductive TCA cycle), glycogen degradation (PWY-5941), pyrimidine biosynthesis (PWY-7187), and vitamin B12-related metabolism (adenosylcobalamin salvage from cobinamide II, PWY-6269). The latter was only found in relation to phenological stages differences.

## 4. Discussion

The main aim of this study was to identify how much the rhizosphere microbiota of potato plants is impacted by the site of cultivation and to examine whether the plant growth stage determines changes in the rhizomicrobiota. Though we are aware of the limitation of a study conducted during only one year and with a limited number of sites, the obtained results clearly indicated that the overall diversity of the microbiota was principally affected by the pedoclimatic conditions of the site of cultivation. However, the plant phenological stage was also shown to play a role in rhizomicrobiota diversity and taxonomic composition. Indeed, the plant developmental stage is a relevant factor influencing rhizosphere microbiota [8]. In a previous study [9], conducted with low-input agriculture in the center of the origin area of potatoes (High Andes), alpha diversity was influenced by the sampling site, while the effects of plant development on alpha diversity were less pronounced. This result is in agreement with evidence from our study, which was conducted using the same variety for all samples (cv. Kennebec). In fact, in our case-study also reported that alpha diversity mostly differs in relation to the locality, and, to a lesser extent, in relation to plant phenological stages. Interestingly, in the same study [9], taxonomic diversity was significantly affected by the phenological stage, which perfectly matches our results, indicating a stronger effect of the phenological stage on the taxonomic structure of the microbiota than that observed for alpha diversity. However, in the case of taxonomic diversity, the main effect was still related to the locality, justifying the possibility of identifying microbiota-related markers for the site of cultivation, as previously observed for other plant species, such as barley, *Crocus,* and orchids [16,17,18].

An abundant core microbiota, constituted by members of phyla Acidobacteriota, Actinobacteriota, Chloroflexi, and Proteobacteria, was found, irrespective of the cultivation site and phenological stage. Acidobacteria, Actinobacteriota, and Proteobacteria were found to be abundant in potatoes in other studies [9,10,42], indicating that differences in varieties, geographical locations, agronomic practices, and laboratory techniques (e.g., DNA extraction and sequencing methods) do not affect the overall taxonomic representation of bacterial potato rhizomicrobiota. Within these main phyla, Actinobacteriota were particularly interesting, since the ASVs related to the among-locality differences were taxonomically affiliated with this group. The presence of bacteria from the phylum Actinobacteriota (mainly from the genus *Streptomyces*) within these ASVs may suggest that the production of secondary metabolites, including novel antibiotics, could be an important function of the potato rhizomicrobiota. In fact, though a species from the genus *Streptomyces*, *S. scabiei*, is the etiological agent of potato common scab disease [43], other strains from the genus *Streptomyces* have shown important antagonistic activities and the suppression of common scab disease [44]. We can envisage a future investigation on *Streptomyces* associated with potato rhizosphere and the isolation of strains with relevant features within this context. Additionally, the presence of ASVs differing by locality also implies the possibility of developing DNA-based markers for such strains to allow for the identification of cultivation sites and the protection of local productions, which is a relevant aspect for traditional Italian food chains [22]. Finally, it is worth noting that the most abundant pathways represented after the PiCRUST analysis of the microbiota were those related to enterobactin biosynthesis and the utilization of phenolic metabolites (p-cymene, p-cumate). Enterobactin is a well known siderophore, helping cells to chelate iron ions and transport them into the cell [45]. Siderophore production is a key function of plant-associated bacteria [46,47] and, in particular, of growth-promoting bacteria [48]. This is highly relevant for bioinoculant formulations, where strains with important, plant growth-promoting functions are combined. Competition for iron is also relevant for bacterial potato pathogens, such as *Dickeya* spp. [49], and a relevant interplay between the plant and the microbiota exists for iron scavenging [50]. We can, consequently, look forward to the isolation of strains with such relevant functions from the potato rhizomicrobiota as ingredients for future bioinoculant formulations to help potatoes with microelement nutrition [51]. The presence of abilities to degrade terpenes is also relevant and highlights the metabolic interaction between the host plant and the microbiota, as observed in other systems [52,53,54,55]. Potatoes—and cv. Kennebec, in particular—are rich in terpene compounds [56], which are present on the skin of tubers and released during growth by the root as the root exudates.

In conclusion, this work reinforces the importance of the site of cultivation for the potato microbiota.It may also suggest the presence of strains and functions that can be further inspected for their possible use in sustainably improving potato cultivation through bioinoculants and developing microbiota-based fingerprinting of potatoes. However, we should be cautious in interpreting present results from a generalized view. In fact, it would be interesting to know how stable this pattern of rhizomicrobiota is in different varieties and throughout the years, especially when farmers change fertilization methods and when different temperatures and different amounts of precipitation occur. Climatic conditions can influence bacterial and fungal pathogens, which, in turn, can influence the overall rhizomicrobiota. Indeed, since sites differed by climatic conditions, additional work should be conducted to evaluate relationships between climatic conditions, rhizomicrobiota, and plant productivity at this smaller geographical scale to better integrate microbiota variation in an agronomic context.

## Figures and Tables

**Figure 1 microorganisms-10-01743-f001:**
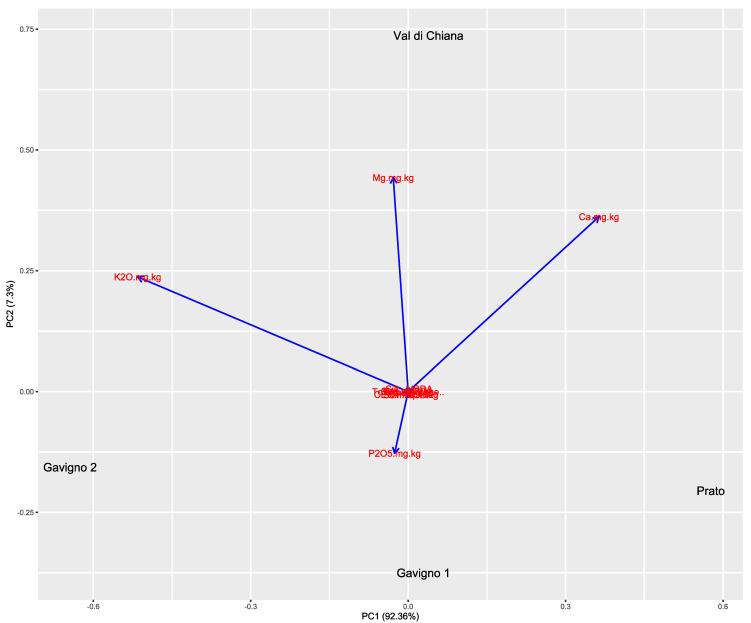
Influence of the physicochemical composition of sampled localities. A principal component analysis, with a biplot, is shown. The percentage of the variance of the first two components is reported. Overlapped labels on the biplot indicate that they do not contribute to the differences among localities.

**Figure 2 microorganisms-10-01743-f002:**
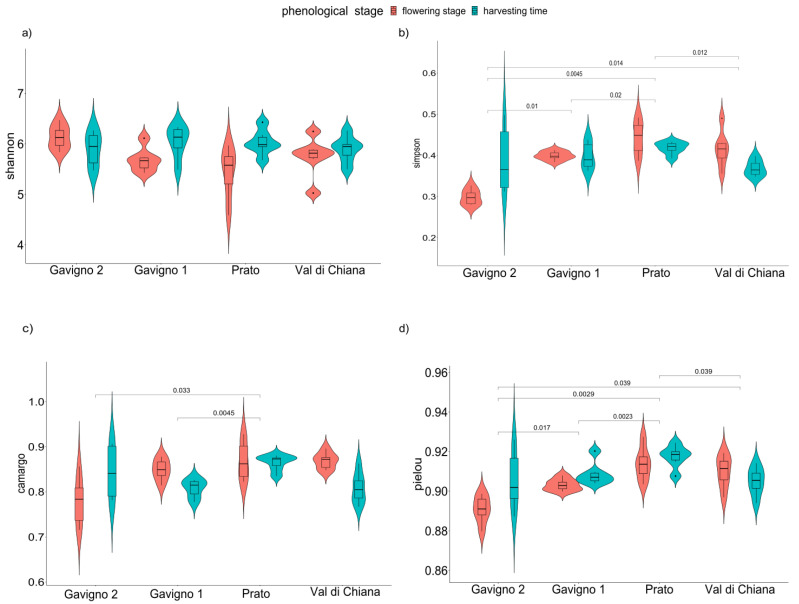
Alpha diversity indices of the samples. (**a**) Shannon index; (**b**) Simpson index; (**c**,**d**) evenness (Camargo‘s and Pielou’s indices, respectively). Lines and the numbers above the lines indicate the *p*-values of the contrasts (Wilcoxon test).

**Figure 3 microorganisms-10-01743-f003:**
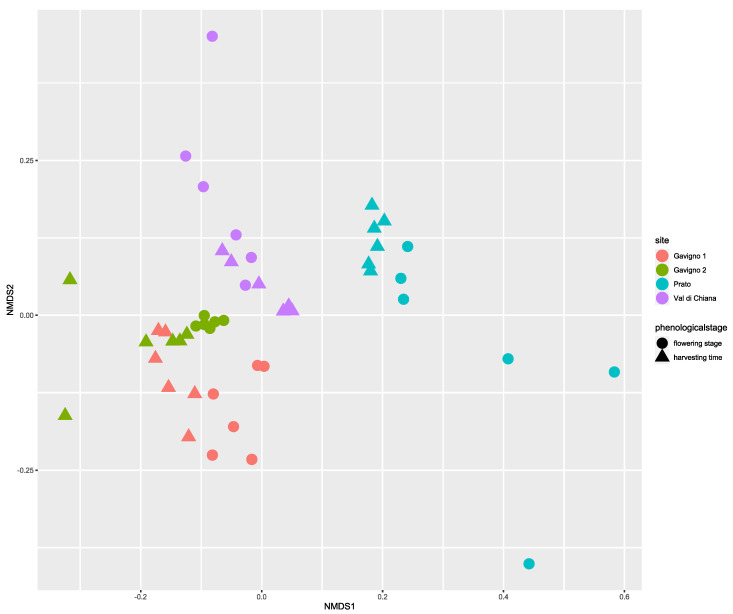
Nonmetric multidimensional scaling of potato bacterial rhizomicrobiota. Colors indicate locations; circles, flowering stages; triangles, harvesting stages. Stress value = 0.17.

**Figure 4 microorganisms-10-01743-f004:**
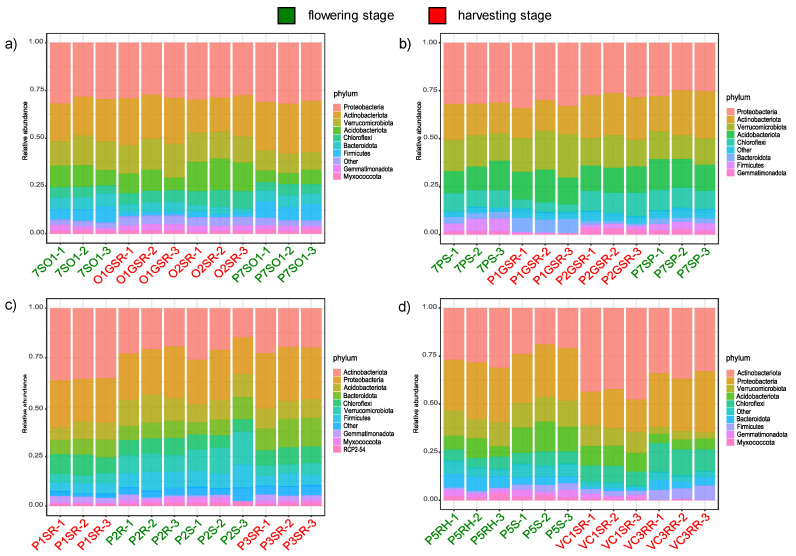
Relative abundances of bacterial phyla in the potato bacterial rhizomicrobiota. The four localities are shown separately. Samples from flowering and harvesting stages are indicated. See Appendix A for sample codes. (**a**) Gavigno 1. (**b**) Gavigno 2. (**c**) Prato. (**d**) Val di Chiana.

**Figure 5 microorganisms-10-01743-f005:**
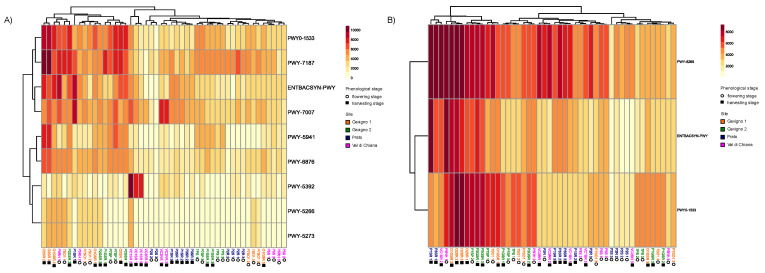
Metabolic pathways mainly contributing to differences among localities (**A**) and phenological stages (**B**). See Appendix A for samples codes.

**Table 1 microorganisms-10-01743-t001:** Sampling sites analyzed in this study. The locality; geographical coordinates; and nitrogen (N), phosphorus (P), and potassium (K) fertilization rates are reported.

Locality	Geographical Coordinates	Altitude	NPK Fertilization Rate
Gavigno 1	44.068862° N, 11.093375° E	739 m	15-5-5 (500 g ha^−1^)
Gavigno 2	44.062105° N, 11.095891° E	763 m	15-5-5 (500 kg ha^−1^)
Val di Chiana	43.412705° N, 11.784870° E	260 m	15-9-15 (100 kg ha^−1^)
Prato	43.860192° N, 11.054822° E	39 m	12-12-17 (350 kg ha^−1^)

**Table 2 microorganisms-10-01743-t002:** Results of PERMANOVAs on alpha diversity indices, showing the contributions of the locality, the phenological stage, and their interaction. Pseudo F-statistic *p*-values are reported. n.s., not significant (*p* > 0.05).

	Shannon		Simpson		Pielou		Camargo	
	F	*p-Value*	F	*p-Value*	F	*p-Value*	F	*p-Value*
Locality	15.064	n.s.	11.9274	0.001	6.2539	0.004	4.4182	0.011
Stage	52.131	0.018	0.4555	n.s.	2.7498	n.s.	0.6968	n.s.
Locality:stage	32.784	0.03	7.5625	0.002	4.7729	0.008	6.7443	0.001

## Data Availability

The sequences dataset was deposited in the SRA database under the BioProject PRJNA730792.

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
