# Peer review of "Effect of Site and Phenological Status on the Potato Bacterial Rhizomicrobiota"

_microorganisms, 2022, doi:10.3390/microorganisms10091743_

Round 1
Reviewer 1 Report
The efforts of the authors in the preparation, editing and overall structuring of the MS are commendable. Though few minor corrections are required:
1. L 29: replace "crops" with 'crop'
2. L 31: replace "panoply" with 'a variety of...'
3. L 52: replace "producer" with 'producers'
4. L 54: insert space "andmountain"
5. L 239: "Here, we wanted" please replace with 'The main aim of this study was to...'
6. L274: please delete '+'
7. L280: "heling cells"??
8. L300: Please avoid using "we"
9. Though the authors have presented a fair account of future prospects of this study but the limitations of this study have only been brushed in L297: "However, we should be cautious to interpret present results on a general view". I strongly recommend all the limitations should be included.
Author Response
We thank the reviewer for the nice comment and having identified errors to be corrected. We fixed all points arisen
Q1. L 29: replace "crops" with 'crop'
A: Done
Q2. L 31: replace "panoply" with 'a variety of...'
A: Done
Q3. L 52: replace "producer" with 'producers'
A: Done
Q4. L 54: insert space "andmountain"
A: Done
Q5. L 239: "Here, we wanted" please replace with 'The main aim of this study was to...'
A: Done
Q6. L274: please delete '+'
A: Done
Q7. L280: "heling cells"??
A: The typo is now corrected (helping)
Q8. L300: Please avoid using “we”
A: Done
Q9. Though the authors have presented a fair account of future prospects of this study but the limitations of this study have only been brushed in L297: "However, we should be cautious to interpret present results on a general view". I strongly recommend all the limitations should be included.
A: We have reported the limitations of the study on lines 298-305. Moreover, an additional sentence has been placed at the beginning of the Discussion section (lines 241-242): “Tough we are aware of the limitation of a study done in one year only and with a limited number of sites,….”
Reviewer 2 Report
Comments on the ms entitled “Effect of site and phenological status on the potato bacterial rhizomicrobiota”.
The reviewed work deals with an important aspect of food production which is the identification and subsequent use of the bacterial rhizomicrobiota. In my opinion, the Authors of the work have achieved their goals, namely: to identify the components of the bacterial rhizospheric microbiota of potato and their changes in relation to the site of cultivation and the plant growth stage. To their credit, the Authors of the paper undertook field research (soil sampling) and paid attention to the climatic and soil conditions at the sites where the potato under study was cultivated. Taking ecological aspects: type of soil, basic weather conditions and phenological stage of plant, into account and using molecular methods yielded good results. The paper identifies factors that affect the quality and quantity of potato bacterial rhizomicrobiota at the sites studied. The Authors also set new goals and sought universal solutions, while supporting traditional cultivation methods used in Italy.

Author Response
We thank the reviewer for the nice comment and having identified errors to be corrected. We fixed all points present in the annotated pdf file